# Semantically Guided Adversarial Testing of Vision Models Using Language Models

## Abstract

In targeted adversarial attacks on vision models, the selection of the target label is a critical yet often overlooked determinant of attack success. This target label corresponds to the class that the attacker aims to force the model to predict. Now, existing strategies typically rely on randomness, model predictions, or static semantic resources, limiting interpretability, reproducibility, or flexibility. This paper proposes a semantics-guided framework for adversarial target selection using the cross-modal knowledge transfer from pretrained language and vision-language models. We evaluate several state-of-the-art models (BERT, TinyLLAMA, and CLIP) as similarity sources to select the most and least semantically related labels with respect to the ground truth, forming best- and worst-case adversarial scenarios. Our experiments on three vision models and five attack methods reveal that these models consistently render practical adversarial targets and surpass static lexical databases, such as WordNet, particularly for distant class relationships. We also observe that static testing of target labels offers a preliminary assessment of the effectiveness of similarity sources, *a priori* testing. Our results corroborate the suitability of pretrained models for constructing interpretable, standardized, and scalable adversarial benchmarks across architectures and datasets.

## 1 Introduction

Over the past decade, a wide range of adversarial attacks have been developed to expose vulnerabilities in deep learning models in computer vision (Szegedy et al., 2013; Goodfellow et al., 2014; Carlini & Wagner, 2017; Byun et al., 2022; Filus & Domańska, 2023). These vulnerabilities pose a distinct but equally crucial security threat alongside traditional security threats (Gelenbe & Nasereddin, 2025; Guerra-Manzanares et al., 2020; Siavvas et al., 2024; Filus & Domańska; Hanif & Maffeis, 2022), by targeting model behavior and knowledge. The absence of simple testing strategies limits the integration of adversarial attacks into established security testing practices, particularly among practitioners handling traditional threats rather than machine learning researchers. Furthermore, the field of adversarial attacks has made significant advancements in devising tweaks that effectively alter models' predictions while remaining imperceptible. As low-level attack techniques have matured, the focus should shift from solely inventing new attack types to systematically evaluating and benchmarking attacks and models, as well as enabling standardized security audits. A critical yet often-disregarded factor is how the target labels are selected for targeted attacks, which can cause severe consequences by deceiving networks into predicting designated target classes (Byun et al., 2022). Most studies either choose random targets or use the predicted classes' probabilities, ignoring the semantic meaning and structure behind label relationships (Kurakin et al., 2016; Carlini & Wagner, 2017; Hu et al., 2021).

The current label-choice strategies are problematic to interpret and explain, which is crucial for testing transparency. For instance, when using the least-likely predicted classes, the selected target labels may depend on unpredictable, image-specific artifacts, such as co-occurring objects or spurious background patterns, or even label noise. Hence, it is unattainable to easily determine why a particular class is considered "least likely" for a given input, nor whether the target is semantically relevant or irrelevant. Since this selection method is tied to the model's current predictions, it varies across individual images, making it hard to verify the feasibility of each target. This lack of transparency and scalability makes image-dependent strategies unsuitable for systematic benchmarking.

Figure 1: Overview of our target label selection method. We leverage semantics from trained language models to guide adversarial testing by identifying the easiest and hardest test cases.

To build meaningful and security-relevant adversarial benchmarks, we argue that target label selection strategies require standardization and should be grounded in interpretable approaches, such as those offered through semantics. Recently, researchers proposed a similarity-driven strategy for target label selection based on class representations embedded in network weights and static language databases (Filus & Domańska, 2024). Selecting targets based on network weights and their perceived class space is beneficial for understanding how a particular network organizes classes and evaluating individual models. However, the internal similarity structure of different models varies, making it unsuitable as a universal reference point for cross-architecture comparison. Therefore, we aim to establish a semantic reference that is agnostic to a target model, facilitating such evaluation. While using the semantics of static lexical databases, such as WordNet, is a step in the right direction, it brings practical and technical disadvantages. WordNet similarity requires manually mapping class labels to specific nodes. While reducing ambiguity, this process is time-consuming, error-prone, and challenging to scale, particularly for datasets with myriad classes. In addition, some parts of WordNet are less refined, which affects similarity metrics based on the normalization of node height (Filus et al., 2025). Moreover, WordNet's tree-like nature leads to discrete and often repeated similarity values, specifically among distant classes. This can cause ambiguity when ranking candidate labels by similarity, which in turn limits the precision of target selection and is dependent on the greedy nature of sorting algorithms. An alternative to static lexical databases could be language and language-vision models, which automatically learn semantic relations. Although purely visual models, language, or multimodal models are trained on different modalities, datasets, and tasks, recent studies suggest a strong alignment between visual and language representations (Ciernik et al., 2024; Filus & Domańska, 2025a; Gao et al., 2022; Radford et al., 2021a; Filus & Domańska, 2025b). This evidence further supports the universality hypothesis, which posits that models trained across diverse scenarios tend to encode concepts comparably, enabling them to represent their respective class spaces in a similar manner (Olah et al., 2020; Chughtai et al., 2023; Kornblith et al., 2019).

We propose a unified framework for semantics-guided target label selection in adversarial attacks, leveraging language and vision-language models to compute semantic similarity between classes. Target classes are selected based on their similarity to the ground truth, with the most similar ones representing the "best-case" targets, those that are easier to reach, and the most dissimilar ones representing the "worst-case" targets, which pose a greater challenge due to their semantic distance. We present a simple overview of this approach in Figure 1. Unlike attacked model-dependent and image-dependent target selection strategies, our similarity-based strategy is reproducible and decoupled from instance-specific confounding factors. It is also interpretable because it is rooted in semantics. This approach allows us to create target lookup tables that are constant throughout evaluation (Filus & Domańska, 2024). As an additional application, we propose the use of the Dissimilarity Metric (DM) according to Filus & Domańska (2023; 2024), which quantifies the shift between labels within the class space defined by the weights of the attacked classifier, as a preliminary tool for assessing the high-level compatibility of different target label sources with the attacked model. In practice, this enables early-stage testing within security audits and the selection of compatible source models, aligning with recent research on image-free testing of image classifiers (Filus & Domańska, 2025a; Nayak et al., 2019; Mopuri et al., 2020; Filus & Domańska, 2025b). To demonstrate the effectiveness of our method, we compare its results with those obtained using WordNet-based similarity and evaluate its performance using standard adversarial attack techniques on a common benchmark dataset, namely the NIPS 2017 Adversarial Competition DEV dataset (Dataset for the NIPS 2017 adversarial competition). We utilize three state-of-the-art text and text-image models as similarity sources, including BERT (Devlin et al., 2019), TinyLLAMA (Zhang et al., 2024), and CLIP (Radford et al., 2021b). We then conduct five attack types on three standard vision models.

Our experiments reveal that using source models from the text domain offers superior results for the worst-case scenario. They also exhibit that text-image models can provide comparable results to WordNet regarding the best-case scenario. The main contributions of this work are: (i) An evaluation of the reliability of text and vision-language models for generating meaningful adversarial targets for vision models. It indicates that knowledge of the models can be effectively used to guide security testing audits in a cross-modal fashion automatically, meaningfully, and transparently. (ii) A comparison between pretrained language models and WordNet in both best- and worst-case scenarios. (iii) An analysis of attack severity based on the model-centric similarity deviation of post-attack predictions from the ground truth labels for different source models. (iv) Evidence that static (dis)similarity computations between source and target models anticipate the behavior of target models under attack. As our proposal is more flexible than static language databases as a similarity source, this shift toward semantically grounded adversarial testing constitutes a crucial step in standardizing security evaluation protocols. It also facilitates consistent and interpretable benchmarking across images, architectures, and datasets, effectively connecting security and explainability.

## 2 RELATED WORK

There is a plethora of works on adversarial attacks in vision primarily focusing on crafting effective perturbations (Li et al., 2024; Madry et al., 2017; Carlini & Wagner, 2017; Szegedy et al., 2013; Moosavi-Dezfooli et al., 2017; Goodfellow et al., 2014; Chen et al., 2024), techniques to obtain transferability between models (Zoph et al., 2018; Zhou et al., 2018; Wang et al., 2023) and attack mitigation (Filus et al., 2024; Bayat & Rish). In contrast, minimal attention has been paid to target label selection in targeted attacks, despite their central role in the interpretability, reproducibility, and standardization of adversarial testing. The most common strategies include choosing random labels (Kurakin et al., 2016; Carlini & Wagner, 2017; Hu et al., 2021) and relying on probabilities of classes predicted for a given image (Kurakin et al., 2018). In the literature, works implement predictions with the lowest or highest probability labels that are not the ground truth, i.e., Least Likely or Most Likely target labels (Kurakin et al., 2018; Hu et al., 2021). Alternatively, the $K$ least or most likely labels can be used (Hu et al., 2021). While these approaches are simple to implement, they inherently depend on instances. This makes the resulting myriad target labels unfeasible to interpret and compare across models. Moreover, probability-based methods are sensitive to factors such as spurious background patterns or object co-occurrence, making it even more unclear why a target label was assigned to a specific image. Such a lack of transparency and scalability makes image-dependent strategies unsuitable for systematic benchmarking.

Moreover, some proposals that connect security and explainability via semantics are still limited. Nevertheless, some works reveal the potential of semantics for adversarial testing. For example, Mahmood and Elhamifar proposed an optimization approach for multi-label learning that modifies the predictions of desired labels while ensuring that other labels are not affected (Mahmood & Elhamifar, 2024). Some authors have considered semantics and other similarity sources to build evaluation metrics in adversarial testing (Mopuri et al., 2020; Filus & Domańska, 2023). While these works reveal the potential of incorporating similarity and semantics for adversarial testing, they focus on different aspects, such as metrics and multi-label optimization. In contrast, we propose to use similarity to guide adversarial testing efficiently and effectively.

A recent work introduced a method for target label selection that exploits similarities derived from internal model representations embedded in weights and lexical databases, such as WordNet, to define best- and worst-case testing scenarios (Filus & Domańska, 2024). In contrast to this work, we propose to transfer the knowledge of trained text and text-image neural networks to guide testing in a cross-modal manner. Our motivation is that recent works in the literature reported about the high alignment between different models and semantics (Ciernik et al., 2024; Filus & Domańska, 2025a; Gao et al., 2022; Radford et al., 2021a; Filus & Domańska, 2025b), with other works suggesting even the universality hypothesis stating that representations of knowledge largely overlap between models trained on different tasks and modalities (Olah et al., 2020; Chughtai et al., 2023; Kornblith et al., 2019). We utilize the embeddings of the known classes, generated with the text-only representation of labels, to estimate similarity between concepts with low computational overhead. We exploit these characteristics to build the best and worst adversarial cases by choosing the most and least similar target labels. This makes our procedure more flexible and more feasible to automate. This is because using WordNet as a source of similarities can be troublesome, as it requires manual mapping of

labels. Besides, there are many equivalent similarities and deficiencies in the scope of certain parts of these databases. Likewise, utilizing target network weights introduces inconsistency in testing when different target models are compared, thereby limiting reproducibility. Hence, this represents the first approach of its kind to adversarial target label selection. We reference the WordNet-based approach presented by Filus & Domańska (2024) as the most adequate baseline for our approach.

## 3 METHODS

### 3.1 SEMANTIC SIMILARITY REFERENCE

Semantic similarity is a relation between terms or objects with a similar meaning. It can be measured, e.g., via static lexical databases with hierarchical, tree-like structures, such as WordNet (Kolb, 2009). The assumption is that the terms to be analyzed are available as the nodes of a given structure. Thus, semantic similarity via WordNet can be measured via the Path Length from Pedersen et al. (2004), WUP from Wu & Palmer (1994), and so forth. In particular, WUP has been widely used to describe semantic similarity between classes (Mopuri et al., 2020; Filus & Domańska, 2024). Hence, we implement this measure to determine our reference best- and worst-case scenarios.

For a given ground truth class $c_{\text{gt}} \in \mathcal{C}$, we define the Most Similar (MS) variant or the best-case scenario class $c^+$ as the class with the highest WUP semantic similarity to $c_{\text{gt}}$, and the Least Similar (LS) variant or the worst-case target $c^-$ as the class with the lowest WUP. These are computed as,

$$c^+ = \underset{c \in \mathcal{C} \setminus \{c_{\text{gt}}\}}{\arg\max} \{\text{WUP}(c_{\text{gt}}, c)\} \quad \text{and} \quad c^- = \underset{c \in \mathcal{C} \setminus \{c_{\text{gt}}\}}{\arg\min} \{\text{WUP}(c_{\text{gt}}, c)\}. \tag{1}$$

We compute $(c^+, c^-)$ pairs for all $c_{\text{gt}} \in \mathcal{C}$ and store them in a lookup table to enable consistent, interpretable, and efficient target label selection during adversarial testing.

### 3.2 SEMANTIC SIMILARITY VIA PRETRAINED MODELS

To compute semantic similarity between classes without relying on static language knowledge bases, such as WordNet, we propose using text-only embeddings computed via different language models (i.e., BERT and TinyLLAMA) and image-language models (i.e., CLIP) trained on large datasets in a self-supervised manner. We refer to these language and image-language models as similarity source models. For each class $c \in \mathcal{C}$, we consider its textual name (e.g., "school bus") and encode it using pretrained models. Let $\mathbf{e}(c) \in \mathbb{R}^d$ denote the resulting embedding for class $c$. Subsequently, the semantic similarity between two classes is assessed using the cosine similarity function, $\text{cs}(\cdot)$. So, given a ground truth class $c_{\text{gt}}$, the MS variant $c^+$ and the LS variant $c^-$ are defined as

$$c^+ = \arg\max_{c \in \mathcal{C} \setminus \{c_{\text{gt}}\}} \{\text{cs}(c_{\text{gt}}, c)\} \quad \text{and} \quad c^- = \arg\min_{c \in \mathcal{C} \setminus \{c_{\text{gt}}\}} \{\text{cs}(c_{\text{gt}}, c)\}. \tag{2}$$

As with the WUP computation, we precompute and store the $(c^+, c^-)$ pairs for all classes using each language model, creating a lookup table (one per source model). This must be done once, before the adversarial training, and therefore does not incur any significant costs during testing. Such an approach supports semantics-driven and consistent testing, thus making security evaluations more interpretable and reproducible. This process avoids manual taxonomy mapping and provides continuous similarity scores, which is the most important for more distant classes. Therefore, the procedure is well-suited for large-scale security evaluations.

### 3.3 ATTACK EVALUATION METRICS

We use two standard metrics to evaluate the success of targeted attacks: Fooling Rate (FR) and Targeted Success Rate (TSR). These are computed for the vision models under attack, namely, the attack target models. FR measures the percentage of images for which the attack caused the label change, regardless of what change it triggered. TSR measures how many targets have been reached by executing the attack. These metrics are mathematically defined as follows,

$$\text{FR} = \frac{1}{N} \sum_{i=1}^{N} \mathbb{1}(\hat{y}_i^{\text{pre}} \neq \hat{y}_i^{\text{post}}) \quad \text{and} \quad \text{TSR} = \frac{1}{N} \sum_{i=1}^{N} \mathbb{1}(\hat{y}_i^{\text{target}} \neq \hat{y}_i^{\text{post}}). \tag{3}$$

In both cases, $\hat{y}_i^k$, $\forall k \in \{\text{pre}, \text{post}, \text{target}\}$, correspond to the $i^{\text{th}}$ pre-attack, post-attack, and target labels, and $N$ is the number of samples in the tested dataset.

We also utilize the Dissimilarity Metric (DM) to measure the severity of attacks from the model's perspective, as described in (Filus & Domańska, 2023). Instead of calculating the attack success on a binary scale, it determines how far the post-attack predicted labels are in the model's perceived space from the ground truth labels. It does that based on the similarity of classes expressed as templates within the target model's final classifier weights. Let $y_i, \hat{y}_i \in \mathcal{Y}$ be the ground truth and predicted label after attack, respectively, for image $i$, since $\mathcal{Y}$ stands for the set of known classes. Let $r(y_i, \hat{y}_i)$ denote the rank distance in the model's similarity space, derived from class template-based similarities within the model's weights. Thus, the overall DM can be determined as follows,

$$\text{DM} = \frac{1}{N} \sum_{i=1}^{N} d_i \quad \text{and} \quad d_i = r(y_i, \hat{y}_i)/(|\mathcal{Y}| - 1), \tag{4}$$

where $|\mathcal{Y}|$ is the classes number, the normalized dissimilarity for one sample is defined as $d_i$. DM takes values in $[0, 1]$, where 0 denotes complete model accuracy, 1 denotes maximal post-attack severity, and intermediate values indicate increasing severity. It corresponds to a mean-normalized rank distance between labels. This metric is beneficial for a more thorough evaluation of attack success than binary TSR, as it estimates the degree of damage caused by the attack, even if the target label was not reached. This metric is crucial for evaluating the LS variant (DM as high as possible) and the MS variant (low, non-zero, with approximately 0.001 being ideal for ImageNet). Using DM aligns with recent efforts to enable model-centric verification procedures (Biecek & Samek, 2025; Filus & Domańska, 2024; 2025b), and with approaches that evaluate their effectiveness beyond simple label flipping (Mopuri et al., 2020), which is still, alarmingly, a widely adopted practice (Zhang et al., 2025; Shao, 2024).

Moreover, we also assessed the values of DM in a "static" setup. This corresponds to the dissimilarity between the ground truth and target values (instead of the predicted values) obtained for different similarity source models, which can be treated as a measure of compatibility between similarity sources. In this case, $\hat{y}_i$ becomes the target label for a given $i^{\text{th}}$ class. Thus, the mean is computed in this setup over all ImageNet classes, not instances in the dataset ($N = |\mathcal{Y}|$). Suppose the standard and dynamic DM results are similar. In that case, it means that we can assess a potential model vulnerability to given targets and similarity sources with no image instances *a priori* to testing.

### 3.4 EXPERIMENTAL SETUP

We conducted experiments with five different attacks: FGSM (Goodfellow et al., 2014), C&W (Carlini & Wagner, 2017), PGD (Madry et al., 2017), Momentum Iterative Method (MIM) (Dong et al., 2018), and Simultaneous Perturbation Stochastic Approximation (SPSA) (Uesato et al., 2018). We utilized three state-of-the-art similarity source models, two from the text domain, BERT (Devlin et al., 2019) and TinyLLAMA (Zhang et al., 2024); and one from the text-image domain, CLIP (Radford et al., 2021b). The models are taken from the Hugging Face repository and implemented in PyTorch. We also employed three vision models as targets for the attacks, such as MobileNetV2 (Sandler et al., 2018), EfficientNetV2B0 (Tan & Le, 2021), and ResNet50V2 (He et al., 2016). The models achieved accuracies of 83.3%, 89.7%, and 84.2%, respectively, on the examined dataset. These models are freely available via the Keras Application and were all trained on a common vision benchmark large-scale dataset, i.e., ImageNet (Deng et al., 2009; Russakovsky et al., 2015). We used the NIPS 2017 Adversarial Competition dataset (Dataset for the NIPS 2017 adversarial competition) for testing, which is a common benchmark employed for adversarial attacks (Byun et al., 2022; Li et al., 2020), and is compatible with ImageNet-trained networks. Moreover, all codes and results of this work can be found in the freely accessible repository at https://github.com/AUTHOR/REPO-ICLR. A portion of the experiments was carried out using a machine from the Grid'5000 testbed with an NVIDIA Tesla P100 GPU with 16 GB RAM, and running on Linux. This is part of the Grid'5000 testbed[1]. The other experiments were performed on a Windows-based workstation with an NVIDIA TITAN RTX GPU and 64 GB RAM.

## 4 EXPERIMENTAL RESULTS

This section presents the results of our experiments designed to assess the effectiveness of semantics-guided target label selection via language models for adversarial vision testing. The analysis is

---

[1]Grid'5000 project is supported by a scientific interest group hosted by Inria and including CNRS, RENATER, several Universities, and other organizations. The authors may not be related to these institutions.

structured around four key research questions: (RQ1) Can text domain models be reliably used to construct best- and worst-case adversarial testing cases? (Section 4.1); (RQ2) How do text and vision-language models compare to static semantic resources like WordNet in selecting target labels for attacks? (Section 4.2); (RQ3) Which similarity sources have the greatest impact on attack severity, as measured by dissimilarity in model predictions? (Section 4.3); (RQ4) Which similarity sources provide targets with the highest static compatibility with vision networks? (Section 4.4). The following subsections present the related results and provide detailed answers.

## 4.1 RELIABILITY OF BUILDING ADVERSARIAL CASES WITH TEXT MODELS

Figure 2 presents the results of different similarity sources for the MS and LS case scenarios. For the MS case (Figure 2a), all similarity sources obtained higher targeted and non-targeted success results than for the LS variant (Figure 2b), which is a desirable outcome. The compact upper-end distributions in the MS violin plots visually support this, indicating more consistent and successful attacks. (Table A.1 in the Appendix A complements Figure 2 with the detailed statistics for FR and TSR per the source model to facilitate analysis.)

For the MS variant, all sources render high FR values (Avg. $\geq 0.7961$), confirming that label changes are easy to induce when the target is semantically close. TSR values, however, vary more. WUP and CLIP achieve the highest average TSR values (0.5563 and 0.5529), followed by BERT and LLAMA (0.4770 and 0.4592), suggesting that WordNet and vision-language models define slightly more accessible best-case targets. Among the similarity sources, CLIP and WUP typically exhibit the lowest variability, implying greater stability and reliability in the test scenarios they produce. In the LS case, FR values are distributed similarly, with Avg. values from 0.7745 to 0.7971, obtained via CLIP and WUP, respectively. In particular, WUP presents the least varied values, with a St. Dev of 0.3495, followed by BERT, which has 0.3759. Notably, the distributions observed in Figure 2b derive their shape from three samples below the 0.5 mark, except for LLAMA, which has two samples, followed by a denser concentration on larger FR values. Additionally, TSR drops markedly across all sources, with means ranging from 0.3919 to 0.4377, which corroborates the increased difficulty of hitting distant semantic targets. CLIP yields the lowest average TSR, followed very closely by LLAMA, indicating that its worst-case selections are maximally challenging. Yet, slightly higher dispersion is observed in the LS variant, especially for CLIP, which reflects more variability in attack success when targets are semantically distant. CLIP and LLAMA exhibit very similar distributions regarding central tendency and dispersion in both FR and TSR. This alignment suggests that, despite their different architectures and modalities, both models capture class-level semantic relationships with considerable accuracy.

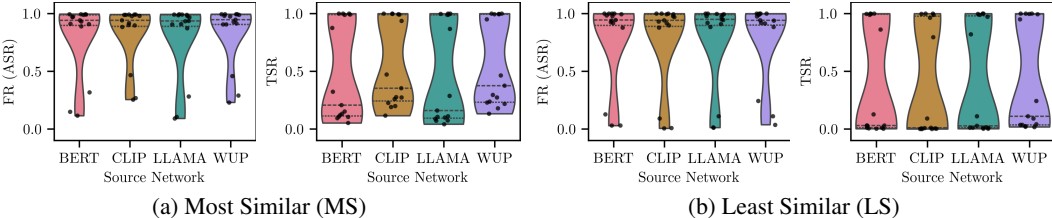

(a) Most Similar (MS)  (b) Least Similar (LS)

Figure 2: Aggregated attack performance for (a) MS and (b) LS target selection strategies across all semantic similarity sources. Fooling Rate (FR) indicates overall label change success, while Targeted Success Rate (TSR) reflects the success in reaching the designated targets. For MS cases, higher values imply easier-to-reach adversarial targets; for LS cases, lower values indicate more challenging ones. Dashed lines denote the quartile ranges, and black dots represent individual samples.

Table 1 provides more detailed MS and LS testing results for the chosen similarity sources. This table also presents the results obtained for WUP, which is our reference. Labels chosen with MS strategies based on all similarity sources are significantly easier to reach than those picked with LS strategies for all target vision models. This is evident in weaker adversarial attacks, such as CW or FGM. For example, the TSR value for EfficientNetV2B0 was 0.3 and 0.04 for MS and LS, respectively. For mighty attacks, PGD in our case, the impact of the label choice is negligible, i.e., the attack success rate oscillates around 100%. These results suggest that the language and image-language models essentially share similar perceptions between classes with pure vision models. Thus, they are good

candidates for target selection and further effectiveness evaluation. While TSR consistently reflects the effectiveness of target label selection, a closer inspection of the FR distributions reveals that label semantics also subtly affect non-targeted success. This is especially evident in the violin's minimum values and lower tails in Figure 2, where MS variants exhibit visibly truncated lower ranges than LS variants. Although the effect on FR is not as pronounced as on TSR, it remains measurable and indicates that semantic proximity plays a role even in less strict attack goals. Such differences, particularly in the bottom quantiles of FR, suggest that label selection influences overall attack difficulty beyond target-hitting accuracy alone.

## 4.2 COMPARING SIMILARITY SOURCES FOR TARGET LABEL SELECTION

We further analyze the results in Figure 2 through Table 1 to determine the use cases where particular similarity sources are better for testing. Text and text-image models produced harder-to-reach LS targets than WUP, as evidenced by lower FR and TSR values. These results support the view that embedding-based similarities better capture global semantic distance. Thus, it aligns with our observation and motivation: WordNet-based similarity measures can lack sufficient variability when they are more distant from the ground truth, due to the presence of many similar values outside the closest neighborhood. Besides, in the case of the MS variant, WordNet-based WUP similarity achieved the best results (Avg. TSR of 0.56), closely followed by the CLIP text-image model (Avg. TSR of 0.55). They also reached the same Avg. FR value. Moreover, BERT and TinyL-LAMA also obtained high FR values (i.e., 0.8

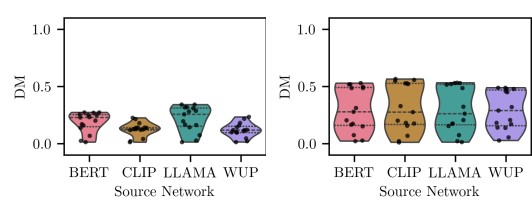

(a) Most Similar (MS)    (b) Least Similar (LS)

Figure 3: Average Dissimilarity Metric (DM) for all similarity sources under (a) MS and (b) LS selection strategies. DM measures the semantic distance between post-attack predictions and ground truth labels in the model's class space. Lower values in (a) and higher values in (b) are desirable and indicate well-aligned similarity assessments. Dashed lines: quartile ranges; black dots: samples.

and 0.81), but the TSR results exhibit that their determined most similar labels are slightly more challenging to reach. Considering the above, the overall results suggest the following practical implications: Text-image models and WordNet-based measures can be slightly better for local similarity estimation. Thus, to construct best-case scenarios for security testing, using text-image models as an alternative to WordNet semantics corresponds to the best approach. Furthermore, text and text-language models are superior at estimating global similarity (i.e., constructing worst-case scenarios).

## 4.3 IMPACT OF SIMILARITY SOURCES ON ATTACK SEVERITY

Figure 3 presents the DM values obtained for a given similarity source in both MS and LS test case scenarios. These values allow us to assess the overall impact of attacks performed with a given target on the model prediction, regardless of the targeted attack's success. Even in the absence of a successful targeted hit, the DM quantifies how far the prediction drifts in the semantic class space. For the MS variant, the average DM value should be low, i.e., labels are perceived as close to the ground truth. Conversely, for the LS variant, high values imply that labels are perceived as distant from the reference. Considering this, the DM results corroborate our finding about the superiority of text and text-image models in selecting the LS labels. Specifically, the best average result obtained for CLIP is approximately 0.31 compared to 0.30 for purely text models and 0.29 for WUP. St. Dev. values are comparable for all source networks. In the case of the MS variant, CLIP again obtained similar results to WUP, whereas the purely text models obtained inferior results with higher dispersion. This shows that WordNet measures and text-image models are more suitable for local similarity estimation. Text and vision-language models are better suited for identifying semantically distant targets, confirming their utility in constructing robust worst-case scenarios.

To support these inferences, Table 2 displays the St. Dev., Min., and Max. values of the DM values. These values are much more variable for the LS testing scenarios. Such a fact is reasonable, as the desired high values in this strategy are much more challenging to reach than those in the MS scenario. The variability, again, is lower for the WUP and CLIP (text-image model) used as similarity

Table 1: Fooling Rate (FR) and Targeted Success Rate (TSR) for Most Similar (MS) and Least Similar (LS) cases across five attack methods (CW, FGM, MIM, PGD, and SPSA), three source models and three target models. Best values per attack-variant pair are in bold.

| Attack | Var. | EfficientNet FR | TSR | MobileNet FR | TSR | ResNet FR | TSR | EfficientNet FR | TSR | MobileNet FR | TSR | ResNet FR | TSR |
|---|---|---|---|---|---|---|---|---|---|---|---|---|---|
| | | | | BERT | | | | | | CLIP | | | |
| CW | MS | 0.15 | 0.15 | 0.32 | 0.32 | 0.12 | 0.12 | 0.27 | 0.28 | **0.47** | **0.47** | **0.26** | **0.26** |
| | LS | 0.03 | 0.03 | 0.13 | 0.13 | 0.03 | 0.03 | **0.01** | **0.01** | **0.09** | **0.09** | **0.01** | **0.01** |
| FGM | MS | 0.89 | 0.13 | **0.93** | 0.09 | 0.91 | 0.05 | 0.88 | **0.27** | 0.92 | **0.19** | 0.90 | 0.12 |
| | LS | **0.88** | **0.01** | **0.94** | **0.00** | **0.92** | **0.00** | **0.88** | **0.01** | **0.94** | **0.00** | **0.92** | **0.00** |
| MIM | MS | 0.99 | 0.99 | 0.99 | **1.00** | 0.97 | 0.88 | 0.99 | 0.99 | 0.99 | **1.00** | **0.98** | 0.94 |
| | LS | 1.00 | 1.00 | 1.00 | 1.00 | 0.98 | 0.86 | 1.00 | 0.97 | 1.00 | 1.00 | 0.98 | 0.80 |
| PGD | MS | **1.00** | **1.00** | 0.99 | **1.00** | 0.99 | 0.99 | 0.99 | **1.00** | 0.99 | **1.00** | **0.99** | **0.99** |
| | LS | **1.00** | **1.00** | **1.00** | **1.00** | **1.00** | 0.99 | **1.00** | **1.00** | **1.00** | **1.00** | **1.00** | **0.99** |
| SPSA | MS | 0.91 | 0.10 | **0.97** | 0.21 | 0.94 | 0.11 | 0.90 | 0.20 | **0.98** | 0.36 | 0.94 | **0.23** |
| | LS | 0.93 | 0.01 | 0.98 | 0.03 | 0.95 | **0.00** | **0.90** | **0.00** | **0.97** | **0.01** | **0.93** | **0.00** |
| | | | | LLAMA | | | | | | WUP | | | |
| CW | MS | 0.10 | 0.11 | 0.28 | 0.29 | 0.09 | 0.10 | **0.29** | **0.30** | 0.46 | 0.46 | 0.23 | 0.24 |
| | LS | **0.01** | **0.01** | 0.11 | 0.11 | **0.01** | **0.01** | 0.04 | 0.04 | 0.24 | 0.24 | 0.11 | 0.11 |
| FGM | MS | 0.87 | 0.10 | **0.93** | 0.07 | 0.91 | 0.04 | **0.91** | **0.27** | **0.93** | 0.18 | **0.91** | **0.13** |
| | LS | 0.89 | **0.01** | 0.95 | **0.00** | **0.92** | **0.00** | **0.88** | 0.02 | **0.94** | 0.04 | **0.92** | 0.02 |
| MIM | MS | 0.99 | 0.99 | 0.99 | **1.00** | 0.98 | 0.87 | **1.00** | **1.00** | **1.00** | **1.00** | **0.98** | **0.95** |
| | LS | 0.99 | 0.97 | 1.00 | 1.00 | 0.98 | 0.82 | 1.00 | 1.00 | 1.00 | 1.00 | 0.99 | 0.95 |
| PGD | MS | 0.99 | **1.00** | 0.99 | **1.00** | 0.99 | 0.99 | **1.00** | **1.00** | **1.00** | **1.00** | **0.99** | **0.99** |
| | LS | **1.00** | **1.00** | **1.00** | **1.00** | **1.00** | 0.99 | **1.00** | **1.00** | **1.00** | **1.00** | **1.00** | **0.99** |
| SPSA | MS | 0.91 | 0.07 | **0.97** | 0.16 | 0.94 | 0.09 | **0.92** | **0.22** | **0.98** | 0.38 | **0.95** | **0.23** |
| | LS | 0.91 | 0.01 | **0.97** | 0.03 | 0.95 | 0.01 | 0.92 | **0.02** | 0.98 | 0.09 | 0.94 | 0.04 |

sources. The minimum values show that even in the LS scenarios, the predicted labels can still be relatively close to the ground truth for some images and weaker attacks. However, the maximum DM values obtained for the LS variant are about 0.5 (0.6 for CLIP), and approximately 0.2/0.3 for the MS variant. This means the LS variant can result in much more distant predictions than the MS one, by about 1/3 of the total rank distance in the model's class space.

Table 2: Statistics of the Dissimilarity Metric (DM) for MS and LS variants. Higher average DM values suggest that attacks caused more severe shifts of labels in the attacked model's class space.

| Statistic | BERT | CLIP | LLAMA | WUP | BERT | CLIP | LLAMA | WUP |
|---|---|---|---|---|---|---|---|---|
| | | Most Similar (MS) | | | | Least Similar (LS) | | |
| Avg. | 0.1881 | 0.1253 | 0.2164 | 0.1197 | 0.2969 | 0.3119 | 0.3027 | 0.2924 |
| St. Dev. | 0.0898 | 0.0613 | 0.1131 | 0.0632 | 0.1933 | 0.2111 | 0.2008 | 0.1710 |
| Min. | 0.0143 | 0.0134 | 0.0148 | 0.0148 | 0.0233 | 0.0124 | 0.0138 | 0.0283 |
| Max. | 0.2739 | 0.2259 | 0.3422 | 0.2339 | 0.5300 | 0.5661 | 0.5334 | 0.4901 |

## 4.4 STATIC COMPATIBILITY OF SOURCE AND TARGET MODELS

Figure 4 presents the static DM scores computed between ground truth labels and target labels selected via different semantic similarity sources, i.e., without using any image data. These scores reflect the mean normalized rank distance from the ground truth in the model's output space. We observe consistent patterns by comparing them with the post-attack DM results in Figure 3. In static and predictive settings, TinyLLAMA produces the highest DM in the MS variant. It indicates that its most similar targets are semantically less aligned than those from other sources. Conversely, CLIP and WUP consistently yield the lowest DM values in the MS case, reflecting better local similarity alignment. These trends are also reflected in FR and TSR from Table 1, where CLIP achieves top performance for MobileNetV2 and ResNet50V2. CLIP again produces the highest static DM values for the LS variant, confirming its effectiveness in identifying worst-case targets. The results suggest

that static DM can assess, *a priori*, the alignment between language similarity sources and vision models' representations. The close match between static and predictive DM trends confirms that it provides an initial compatibility measure for evaluating and selecting semantic sources.

# 5 DISCUSSION

Our experimental results show that language models can be effectively used to construct best- and worst-case scenarios for adversarial tests of vision models. Best-case scenarios are valuable for model-centric tests, as they should be robust even against easy targets, such as "child" for "adult". In contrast, worst-case scenarios are functional for attack-centric tests, as they should be stealthy and successful even against challenging targets, such as "tree" for "person". An apparent limitation is that language models are less interpretable than static lexical resources. However, recent advances in this direction, primarily via mechanistic methods such as Sparse Autoencoders (SAEs), facilitate the identification of disentangled concepts from which networks construct classes. Hence, our method can be enhanced with SAE expla-

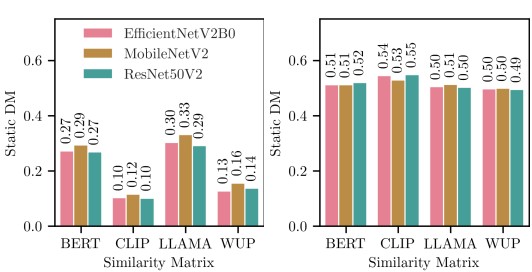

(a) Most Similar (MS)    (b) Least Similar (LS)

Figure 4: Static Dissimilarity Metric (DM) computed for all ImageNet classes, based on MS and LS target labels obtained from different semantic similarity sources.

nations. We also aim to explore it in future work. Plus, one might argue that the DM computation could leverage similarity estimations more fully than relying only on similarity via classifier weights, as in the basic variant (Filus & Domańska, 2023; 2024). Nevertheless, numerous studies have demonstrated that this estimation is sufficient for multiple practical applications (Nayak et al., 2019; Mopuri et al., 2020; Filus & Domańska, 2025a; 2024). Since our scope is on the complete attacked model operation, this formulation of DM remains both justified and effective.

# 6 CONCLUSIONS

In this work, we proposed a unified framework for semantics-guided adversarial target label selection that leverages language and mixed models to estimate semantic similarity between class labels. This approach enables the construction of interpretable and systematic best- and worst-case adversarial scenarios. Unlike attacked model- or image-dependent methods, our strategy offers a reproducible and scalable alternative that decouples target selection from instance-level confounding factors. Our experimental evaluation, which covered three similarity source models, three vision architectures, and five attack methods, demonstrated that pretrained text and text-image models are highly effective in guiding adversarial target selection via both best- and worst-case scenarios. The text-image model CLIP consistently outperformed the WordNet-based baseline in identifying semantically distant and hard-to-reach labels, and performed on par with it in the best-case scenarios. Additionally, the DM applied to post-attack predictions revealed that text and vision language embeddings provide a more accurate estimation of global class relationships. At the same time, CLIP and WordNet remain competitive in capturing local similarities. A noteworthy insight from our analysis is that the static DM, computed without requiring image inputs, can serve as an early indicator of how compatible a given similarity source is with a vision model's internal class perception. The strong alignment between static and predictive metric trends suggests that model vulnerability can, to some extent, be assessed in advance, enabling proactive security diagnostics before testing begins. This thrilling observation will be utilized in future work to investigate how perturbations are transferred between networks based on their high-level compatibility. Our findings support the use of semantics-aware target selection via pretrained models as a practical, flexible, and effective alternative to static lexical databases. Our method improves transparency and standardization in adversarial evaluation, laying the groundwork for more reproducible testing pipelines and interpretable security audits. We aim to extend this framework to adapt contextual and image-conditioned semantics, as well as adjust it for multi-label and open vocabulary classification.

ETHICS STATEMENT

This work investigates the use of language/language-vision models to identify test cases for adversarial attacks on vision models with the aim of advancing robustness research. All experiments are conducted on publicly available datasets and models, and no sensitive data are used. The methods developed are intended solely for scientific study and should not be applied for malicious purposes.

REPRODUCIBILITY STATEMENT

All codes and raw experimental results, in CSV format, will be made publicly available via our GitHub repository upon publication. For the review process, we have provided the complete repository as supplementary materials. Our experiments rely exclusively on freely available models from the Keras Applications library, ensuring that all reported results can be reproduced without restrictions and employing well-established frameworks.

ACKNOWLEDGMENTS

This work was supported by XXX XXXXXXX XXXXXX XXXXX XX XXX XXXXXXXXXX XXXXXXXXX XXXXXXXXXX XXXXXXXXXX XXXXXXXX XXXXXXXX, XXXXXXXXX XXXX XXXXXXX XXXXXXXXX XXXXXXXX XXXXXXXX XXXXXXXX.

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

## A    NUMERICAL RESULTS FOR BINARY ATTACK METRICS

To facilitate the analysis of Figure 3, we provide the numerical values of basic statistics of the obtained Fooling Rate (FR) and Targeted Success Rate (TSR) results in Table A.1. As discussed in the main paper, the MS variant renders significantly higher TSRs and FRs than the LS variant, indicating that our best-case variant is more easily attainable than our worst-case variant. This has a beneficial practical implication: the MS variant can be used in model-centric robustness tests, as its targets are easily reachable. Thus, models should be robust enough to deal with even such targets. The LS variant, on the other hand, is functional when new attacks are tested, because a stealthy attack should be able to reach even difficult labels. This can result in serious consequences in practical applications.

Table A.1: Statistics of FR and TSR for Most Similar (MS) and Least Similar (LS) variants. For MS cases, higher values imply easier-to-reach adversarial targets, while for LS cases, lower values indicate more challenging targets. The best results per row are highlighted in bold across the sources.

| Variant | Statistic | Fooling Rate (FR) | | | | Targeted Success Rate (TSR) | | | |
|---|---|---|---|---|---|---|---|---|---|
| | | **BERT** | **CLIP** | **LLAMA** | **WUP** | **BERT** | **CLIP** | **LLAMA** | **WUP** |
| MS | Avg. | 0.8052 | 0.8302 | 0.7961 | **0.8343** | 0.4770 | 0.5529 | 0.4592 | **0.5563** |
| | St. Dev. | 0.3209 | **0.2653** | 0.3342 | 0.2684 | 0.4274 | 0.3757 | 0.4402 | **0.3745** |
| | Min. | 0.1154 | **0.2557** | 0.0916 | 0.2308 | 0.0520 | 0.1154 | 0.0419 | **0.1324** |
| | Max. | **0.9966** | 0.9932 | 0.9943 | 0.9955 | **1.0000** | **1.0000** | **1.0000** | **1.0000** |
| LS | Avg. | 0.7837 | **0.7745** | 0.7790 | 0.7971 | 0.4066 | **0.3919** | 0.3989 | 0.4377 |
| | St. Dev. | 0.3759 | 0.3851 | 0.3818 | **0.3495** | 0.4825 | 0.4816 | 0.4799 | **0.4704** |
| | Min. | 0.0294 | **0.0057** | 0.0124 | 0.0373 | 0.0023 | **0.0000** | 0.0023 | 0.0181 |
| | Max. | **1.0000** | **1.0000** | **1.0000** | **1.0000** | **1.0000** | **1.0000** | **1.0000** | **1.0000** |

