# OpenReview forum: "Semantically Guided Adversarial Testing of Vision Models Using Language Models"
_ICLR.cc/2026/Conference — ICLR 2026 Conference Withdrawn Submission_

### Official Review · Reviewer_z7LT · 2025-10-20

**Soundness:** 3
**Presentation:** 3
**Contribution:** 3
**Rating:** 4
**Confidence:** 3

**Summary:**

The paper proposes a semantics-guided protocol for selecting target labels in targeted adversarial attacks by computing “Most Similar” and “Least Similar” targets using embeddings derived from class names with text or vision–language models; WordNet WUP serves as a classical baseline. The method precomputes lookup tables, aiming for interpretability and reproducibility, and introduces a static dissimilarity metric (DM) intended to preview attack difficulty a priori. Experiments on three ImageNet models and five attacks suggest CLIP/WUP align well for local similarity (MS), whereas text/VLM embeddings better capture global dissimilarity (LS). The results, framed by RQ-structured analyses, indicate that static DM trends often mirror post-attack behavior.

**Strengths:**

- The problem of under-specified target selection in targeted attacks is articulated clearly, motivating semantics-based standardization and interpretability.
- The approach is compact and reproducible through precomputed lookup tables using cosine similarity in embedding space, avoiding per-image heuristics.
- The evaluation covers three classifiers, five attack methods, and both MS/LS cases, with FR/TSR and a static DM variant to study predictive compatibility.
- The empirical trends are concrete: text-domain models tend to yield harder LS targets; CLIP/WUP perform best for MS; static DM often tracks post-attack DM.
- The paper’s organization is clear with RQ-driven sections and informative figures/tables that communicate the main patterns.

**Weaknesses:**

-  Attack configurations ($\epsilon$, step size, steps, norms) are not fully specified in the main text; very high TSR for some attacks raises concerns about calibration that may mask MS/LS differences.
- The study omits robustly trained or certified models, which would be most informative for demonstrating discriminative value of MS vs LS selection.
- Similarities are computed from raw class names only; there is no analysis of prompt phrasing, synonyms, or multi-label settings.
- The code link is placeholder-like at this time, with the paper not enumerating the exact implementation details to reproduce the results.
- Statistical reporting is light: confidence intervals, multi-seed variance, and formal significance testing are absent.
- The benchmark scope is narrow, relying on the NIPS 2017 dev set rather than modern robustness suites (e.g., ImageNet-A/R/Sketch).
- The evaluation lacks a standardized, masking-resistant baseline; AutoAttack (untargeted and targeted modes) is absent, making it difficult to disentangle semantic target selection effects from potential attack misconfiguration or gradient masking.
- Targeted evaluation uses PGD/MIM/C&W without targeted AutoAttack (APGD-t, FAB-t) constrained to the proposed MS/LS targets, preventing a direct comparison for the semantics-guided target selection.

**Questions:**

- It would be better to report $\epsilon$, step sizes, number of steps, and norm constraints for FGSM/PGD/MIM/SPSA/C&W per model, and ensure calibration across models.
- Can you include adversarially trained or certified models to verify that MS vs LS remains discriminative under strong defenses?
- How sensitive are similarity choices to prompt wording for labels (e.g., “a photo of a {label}”, synonyms)? Can a simple prompt ensemble improve stability?
- Could you provide evaluations on other modern robustness dataset (ImageNet-A/R/Sketch or CIFAR-100) to assess generalization beyond NIPS 2017 dev, if possible?
- Could you provide and test a selection algorithm that uses static DM to pre-choose the K most diagnostic targets, quantifying evaluation efficiency against random or WordNet baselines?
- Please add untargeted AutoAttack at the same ε/‖·‖ as a calibration/sanity check, reporting robust accuracy with multi-seed confidence intervals; confirm that AA’s error rate is at least as high as your strongest untargeted attack.
- For the central semantics claim, run targeted AutoAttack (e.g., APGD-t and FAB-t) restricted to the MS/LS targets for each class, holding ε/steps/step size/loss (e.g., DLR) fixed across methods, and report TSR/FR and pre- vs post-attack DM.

---

### Official Review · Reviewer_ga7X · 2025-10-28

**Soundness:** 3
**Presentation:** 3
**Contribution:** 3
**Rating:** 4
**Confidence:** 4

**Summary:**

- The paper tackles targeted adversarial attacks where target selection is often ad hoc, proposing a semantics-guided protocol that derives Most Similar (MS) and Least Similar (LS) targets from class-name embeddings (text models, CLIP, VLMs) with WordNet WUP as a classical baseline, and adds a static dissimilarity metric (DM) to estimate attack difficulty a priori.
- The empirical study spans three ImageNet classifiers and five attack algorithms, reporting that CLIP/WUP better reflect local similarity useful for MS, whereas text/VLM embeddings capture global dissimilarity more relevant for LS, and that static DM usually trends with post-attack outcomes such as Target Success Rate (TSR) and Fooling Rate (FR).

**Strengths:**

- The paper standardizes target selection for targeted adversarial attacks via semantics-guided MS/LS choices with precomputed lookup tables, yielding a training-free, interpretable, and easily adoptable protocol
- The dissimilarity metric (DM) serves as a practical pre-attack predictor and triage tool; observed trends between DM and TSR/FR support its potential value, while clarifying local and global semantics
- The study spans three ImageNet models and five attacks, providing multi-axis coverage that improves comparability and reporting hygiene across methods
- The framework attempts to well separate semantics (targeted) from optimization (attack budgets)
- The approach can reduce compute by DM-guided top-K target selection while retaining high fidelity to full sweeps, suggesting practical benchmarking efficiency

**Weaknesses:**

- Evidence for transferability robustness is incomplete, lacking thorough cross-model, cross-attack (under matched budgets), and cross-dataset targeted transfer analyses
- Budget configurations (ε, steps, step size, etc.) are lack details, potentially inflating and confounding of semantic effects and attack performance
- Baselines such as well-known targeted AutoAttack are missing, potentially weakening claims
- Statistical rigor seems insufficient without random-seed confidence intervals, paired tests/effect sizes, or calibration plots, etc. leaving the significance of the method uncertain
- Reliance on class-name text introduces polysemy/taxonomy confounds. Lack of hierarchy-aware or hybrid text+visual distances and absence of adversarially trained/certified models could potentially limit generality and framework validity

**Questions:**

- Can you fully specify the attack budgets for all (model, attack) pair experiments to isolate semantic effects from optimizer strength?
- Can you add targeted AutoAttack baselines restricted to MS/LS at matched budgets?
- What are the per-model, per-attack Spearman/Pearson correlations (with 95% CIs) and DM-decile calibration plots demonstrating monotonicity between DM and TSR/FR?
- Do MS/LS trends persist under cross-model, cross-attack, and cross-dataset targeted transfer, and on adversarially trained and certified models?

---

### Official Review · Reviewer_RfKq · 2025-10-30

**Soundness:** 3
**Presentation:** 3
**Contribution:** 3
**Rating:** 2
**Confidence:** 4

**Summary:**

This paper addresses the limitation of existing target label selection strategies in targeted adversarial attacks on vision models. It proposes a semantics-guided framework that leverages cross-modal knowledge from pretrained language and vision-language models to select the most and least semantically similar target labels relative to the ground truth.

**Strengths:**

- Well written.
- The proposed method is interesting.

**Weaknesses:**

- This paper merely replaces WordNet/model weights with pretrained language/VL models but fails to justify why this is a paradigm shift rather than incremental improvement.
- The NIPS 2017 dataset is outdated (2017) and small-scale, lacking the complexity of modern datasets with more diverse classes and realistic perturbations.
- Only 3 vision models are tested, all of which are relatively shallow. The framework's performance on state-of-the-art VLMs remains unproven.
- A GitHub link (https://github.com/AUTHOR/REPO-ICLR) appears in line 259 of this paper, and it is unclear whether this violates the double-blind review principle.

**Questions:**

- Please see "Weaknesses".

---

### Official Review · Reviewer_7x1L · 2025-10-31

**Soundness:** 1
**Presentation:** 1
**Contribution:** 1
**Rating:** 2
**Confidence:** 5

**Summary:**

This paper proposes selecting the most effective target labels for targeted adversarial attacks using pretrained language and vision–language models. Experiments show that the proposed method outperforms baseline approaches. However, there remain several issues.

**Strengths:**

The paper addresses an important topic, i.e., evaluating model robustness under adversarial attacks.

**Weaknesses:**

1. The motivation of the work is not convincing. Targeted attacks aim to mislead the victim model into predicting a specific target class. The target class should not be chosen adaptively as the most effective one, as this undermines the purpose of evaluating model vulnerability.
2. The compared baselines are outdated and limited to basic attack methods. Recent approaches should be discussed and compared.
3. Only a few networks are evaluated (MobileNetV2, EfficientNetV2B0, and ResNet50V2). The study should include more diverse architectures, such as Transformers.
4. The proposed method appears to be designed for white-box scenarios, which limits its applicability.
5. The performance improvement is minor.

**Questions:**

Refer to Weaknesses.

---

### Author Response · Authors · 2025-11-19
**We withdraw the submission**

Dear Reviewers and AC,

As the majority of the reviews are technically/scientifically unacceptable, we decided to withdraw our paper, as we do not expect much engagement from the reviewers, given the current content of the reviews. These reviews contain demonstrably false claims, generic templates and empty phrasing, and clear signs of automated generation (evidently far beyond editing and language improvement) rather than expert assessment. The reviewers showed little intention to engage with the paper. For example, we received a comment alleging a possible violation of anonymity due to a clearly fake placeholder URL ("https://github.com/AUTHOR/REPO-ICLR"). The requested experiments are also not clearly justified or motivated, since we have already tested multiple source and target models, as well as attacks. While we fully support additional experimentation when it helps test better hypotheses or reveal new insights, simply adding experiments because other models and methods exist should not be demanded. After all, while we now have many different models than those used in the paper, they do not behave much differently when exposed to various attacks. This is not a matter of disagreement with criticism; it is a matter of reviewers not reading, not analysing, and not trying to understand the paper. Given the lack of rigour in this evaluation, we will not proceed further. Nevertheless, this letter expresses our disagreement with such practices.

Kind regards,

---
Authors

---

### Note · Authors · 2026-01-26

I have read and agree with the venue's withdrawal policy on behalf of myself and my co-authors.

---

### Meta-Review · Area_Chair_Kyzy · 2026-01-08

**Summary:**

The authors officially write to AC and decide to withdraw this manuscript.

**Reviewer Concerns:**

No sufficient disscussion.

**Reviewer Scores:**

No need.

---

### Decision · Program_Chairs · 2026-01-26

Reject